# Multi-Ported GC-eDRAM Bitcell with Dynamic Port Configuration and Refresh Mechanism †

Roman Golman [1,*], Robert Giterman [2] and Adam Teman [1]

1   Emerging Nanoscaled Integrated Circuits and Systems (EnICS) Labs, Faculty of Engineering,
    Bar-Ilan University, Ramat Gan 5290002, Israel; adam.teman@biu.ac.il
2   Telecommunications Circuits Laboratory (TCL), École Polytechnique Fédérale de Lausanne (EPFL),
    1015 Lausanne, Switzerland; robert.giterman@epfl.ch
*   Correspondence: xfory2@gmail.com; Tel.: +972-54-587-2406
†   This paper is an extended version of our paper published in IEEE International Conference on Electronics,
    Circuits, and Systems.

**Abstract:** Embedded memories occupy an increasingly dominant part of the area and power budgets of modern systems-on-chips (SoCs). Multi-ported embedded memories, commonly used by media SoCs and graphical processing units, occupy even more area and consume higher power due to larger memory bitcells. Gain-cell eDRAM is a high-density alternative for multi-ported operation with a small silicon footprint. However, conventional gain-cell memories have limited data availability, as they require periodic refresh operations to maintain their data. In this paper, we propose a novel multi-ported gain-cell design, which provides up-to *N* read ports and *M* independent write ports (*NRMW*). In addition, the proposed design features a configurable mode of operation, supporting a hidden refresh mechanism for improved memory availability, as well as a novel opportunistic refresh port approach. An 8kbit memory macro was implemented using a four-transistor bitcell with four ports (2R2W) in a 28 nm FD-SOI technology, offering up-to a 3× reduction in bitcell area compared to other dual-ported SRAM memory options, while also providing 100% memory availability, as opposed to conventional dynamic memories, which are hindered by limited availability.

**Keywords:** two-port; dual-port; multi-port; 2R2W; 6R2W; N-ported; GC-eDRAM; refresh; configurable memory; 1R1W

## 1. Introduction

As CMOS technology continues to scale, high density embedded memories are of great interest for modern microprocessors and other VLSI system-on-chip (SoC) designs. In fact, embedded memories often dominate the total area and power budget of entire SoCs [1]. Various blocks that are integrated on these SoCs, such as media, graphics processing units, and computational cores, require multi-ported embedded memories to improve their bandwidth by enabling multiple operations using the same memory bank in parallel. For example, the dual-port memory is well utilized for video processing units because several read and write access operations are possible at the same time [2,3]. A larger number of read ports is also utilized in graphics processors, in machine learning applications [4], and as shared system level memory that provides simultaneous access to multiple devices [5,6]. A large number of write ports can be found in multi-threaded applications [7] and vector processors. These, and many other applications and systems, raise the need for *N*-ported memory design [8–13].

The majority of conventional embedded memories are based on the single-ported six-transistor (6T) static random access memory (SRAM) bitcell, shown in Figure 1a. This cell is popular thanks to its symmetrical structure, robust operation, and quick generation using dedicated SRAM compilers. The 6T bitcell utilizes a pair of cross-coupled inverters (M1/M3 and M4/M6) for static data storage and provides read/write access through a

pair of access transistors (M2 and M5). This pair of access devices supports either a single read or a single write operation during a given clock cycle, and therefore it is considered to have a single port, also denoted as "1RW" [3]. While architectural techniques, such as array duplication, can be applied to provide more than one read or write per clock cycle, larger bitcells are required to achieve true multi-ported functionality, resulting in significant area and power overheads [13,14]. Commonly found examples of such bitcells include the eight-transistor dual-ported cell of Figure 1b, which has two pairs of access transistors to provide two completely independent read and write operations (2RW); the eight-transistor two-ported cell, shown in Figure 1c, that comprises a decoupled read port (M7/M8) to enable simultaneous read and write accesses (1R1W) [3]; and the ten-transistor cell, shown in Figure 1d, which has two decoupled read ports that can be used to enable two reads and one write (2R1W) in a parallel or a single differential read [2].

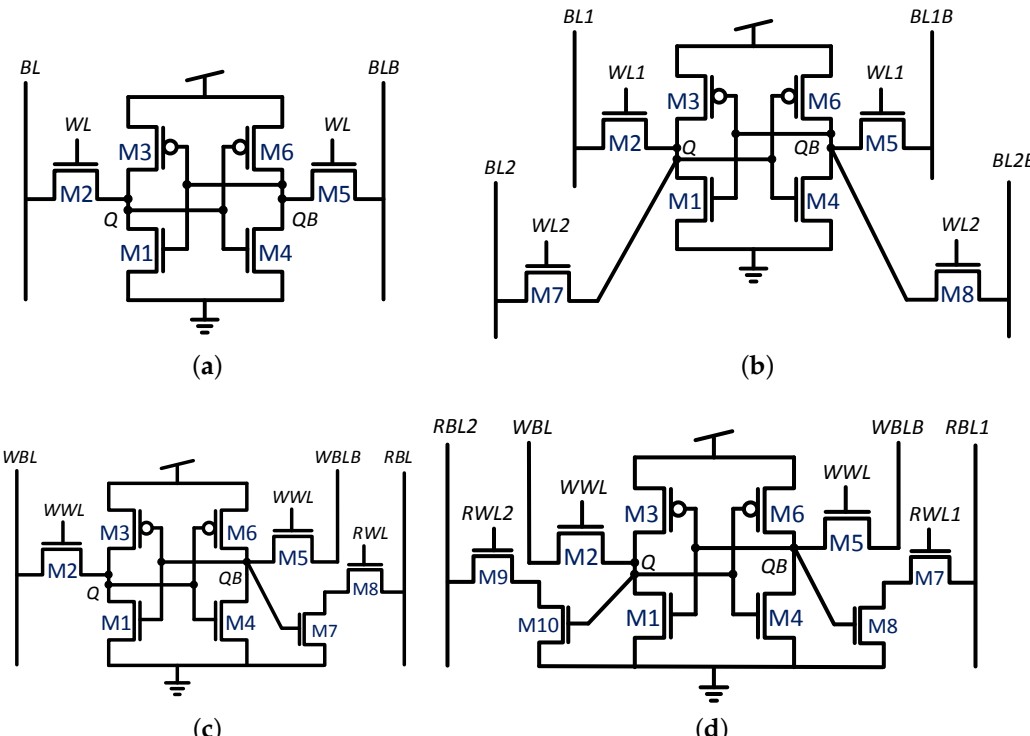

**Figure 1.** Commonly found single and multi-ported SRAM bitcells [15]. (**a**) 6T single-port bitcell; (**b**) 8T dual-port (2RW) bitcell; (**c**) 8T two-port (1R1W) bitcell; (**d**) 10T three-port (2R1W) bitcell.

Examples of bitcells with a higher number of ports include the 24-transistor SRAM bitcell of [6] and the 16-transistor bitcell of [5]. The former, illustrated in Figure 2a, provides 6R2W functionality, achieving improved read stability by adding drivers between Q/QB and the read ports to improve the read static noise margin (SNM). The latter, illustrated in Figure 2b, provides 5R1W functionality using several dedicated read ports. Another recently suggested SRAM-based bitcell is the 8R1W 20T bitcell [4] targeting deep learning applications, illustrated in Figure 2c. This bitcell has a separated differential write port, two drain-connected single transistor read ports, and six gate-connected read ports with two transistors. Half of the read ports are connected to Q and the other half are connected to QB. The inverted read can be flipped by single inverter connected to the column output. Another suggested approach is performing double-pumped read and write operations as suggested in [12,16]. For example, applying this technique to a 3R3W bitcell in [16] transforms it into a 6R6W bitcell. While this technique significantly reduces the bitcell area and macro routing due to a lower number of word-lines and bit-lines, it requires a complex control system and an internal Clock Pulse Generator (CPG). Additionally, the cell requires fast read and write operations, adding additional design constraints. A bitcell suggested for

double pumping is illustrated in Figure 3. However, beyond these, bitcells with more than two read ports are almost unaddressed and can hardly be found in the literature, and the SRAM bitcells, presented in [4–6], come at a high cost in area and often result in decreased SNMs as a result of the additional read ports. Another approach to multiported memory design is standard cell-based memory which utilizes Data Flip-Flop (DFF) or D-latch cells as memory elements [17]. While this technique has low design costs, it comes at the cost of very large area and power overhead [13,18,19].

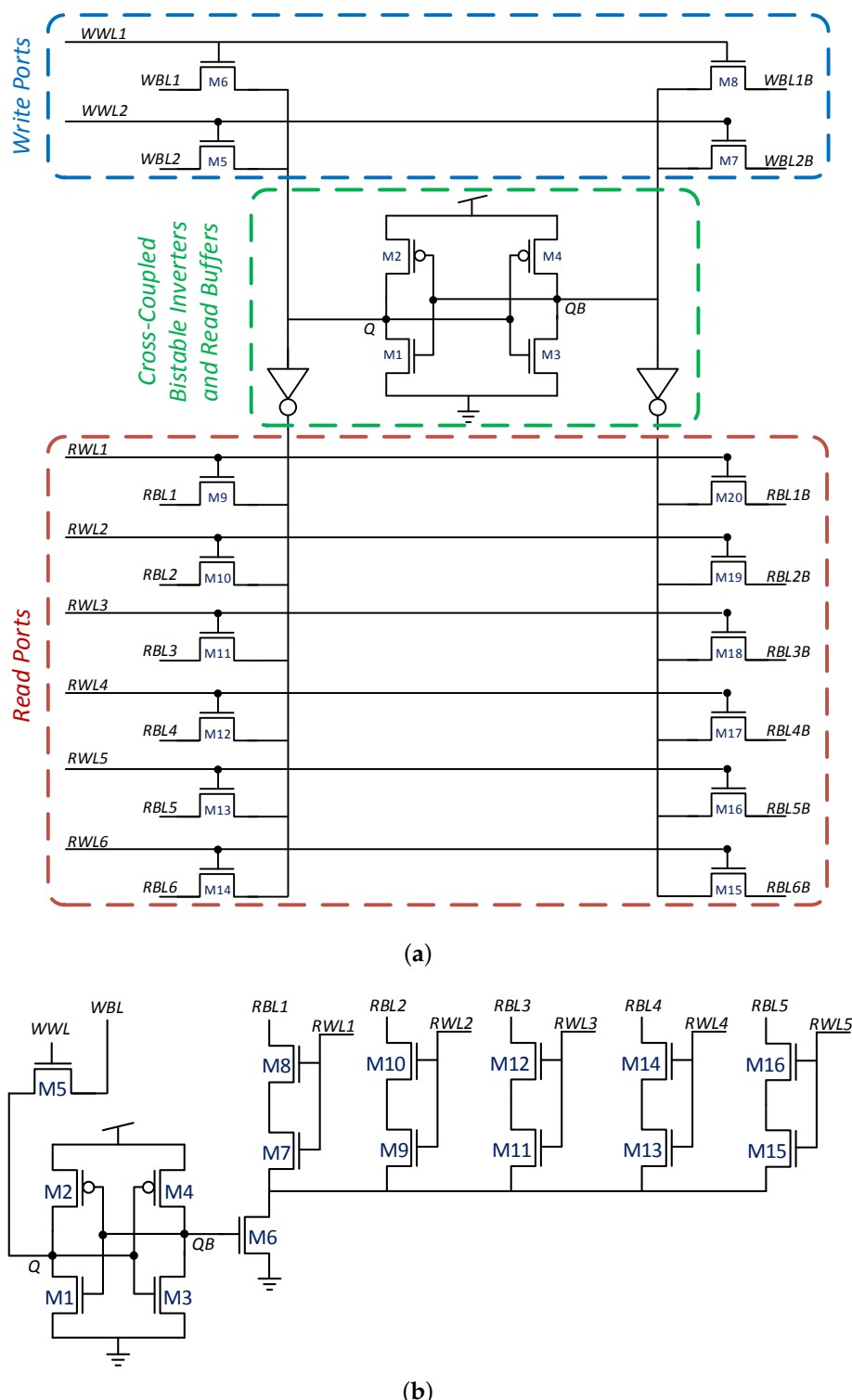

**Figure 2.** *Cont.*

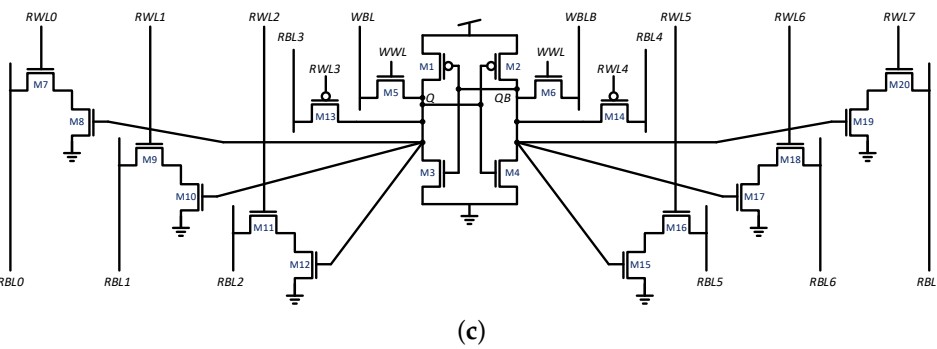

**(c)**

**Figure 2.** Multi-ported SRAM bitcells with a large number of read ports, based on single ended and differential read schemes. (**a**) 24-transistor 6R2W SRAM Bitcell [6]; (**b**) 16-transistor 5R1W SRAM Bitcell [5]; (**c**) 20-transistor 8R1W SRAM Bitcell [4].

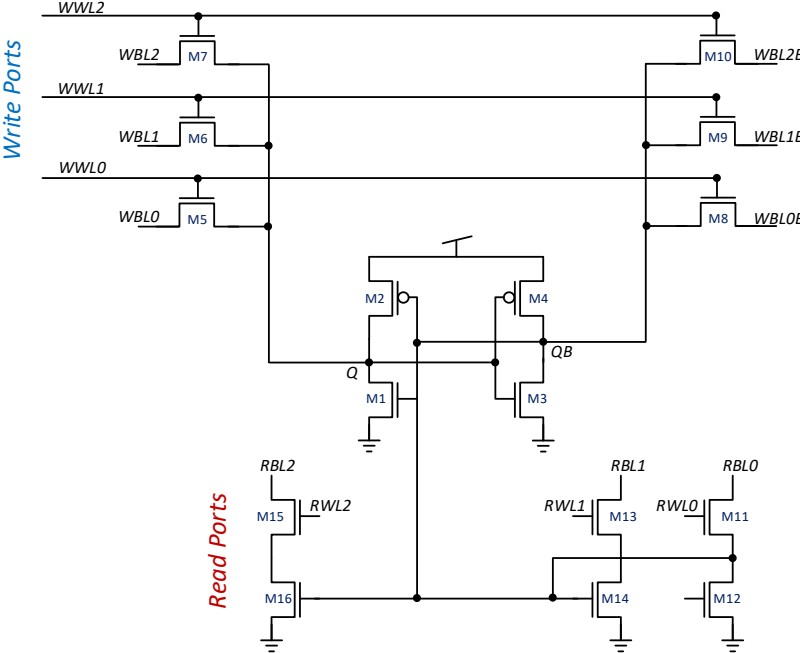

**Figure 3.** 16-transistor 3W3R bitcell which is double pumped to 6R6W functionality in [16].

Another actively researched approach in many core computing applications is utilizing the system interconnect for storing the data in local single ported caches to improve the memory accessibility. While this approach provides throughput improvement up to 2.5× [20,21], it requires complex cache coherency management and large area overhead. An additional similar approach is near memory computing that places the memory macros near the computational units, both reducing the cost of driving the data to the module and also allowing each computational unit to have exclusive access to the data it requires. This approach is very compatible with machine learning (ML) and data encryption and decryption operations [22]. Often, such works embed processing elements within the memory to achieve what is commonly referred to as "in-memory computing" [23–25]. Memories constructed with emerging devices, such as resistive and magnetic RAM, are commonly used for this purpose [26–28]. An example of such a work is described in [29], where highly efficient storage of model weights and personalized data is securely achieved using such memories. Interfacing with these dedicated memories very often requires multi-ported functionality [30–32].

As a high-density alternative with inherent two-ported functionality, gain-cell embedded DRAM (GC-eDRAM) has become increasingly popular in recent years [33–38]. In addition to providing their 1R1W functionality, these memories provide a non-destructive

read operation and low leakage power within a small silicon footprint. Moreover, compared to conventional 1T-1C eDRAM, gain cells are fully logic-compatible, not requiring any non-standard process steps to fabricate the cell capacitance. Recent work has focused on using GC-eDRAM as an area-efficient storage element for embedded in-memory computing applications [39–43] However, as a dynamic memory, periodic refresh operations are required for data retention, limiting the memory availability.

In this paper, we propose a novel GC-eDRAM bitcell topology constructed with $(N + M)$ transistors ($[N + M]$T) to simultaneously provide up to $N$ independent reads and $M$ independent writes ($NRMW$). The proposed memory has several modes of operation: $NRMW$ mode provides $(N + M)$-ported functionality at a very low area and power cost, utilizing a single transistor as a write or read port; and $[N − 1]$R$[M − 1]$W mode, which can be used to hide refresh operations from the outer architecture to provide 100% memory availability. As such, the proposed design provides an area-efficient solution for the realization of multi-ported memories, such as digital signal processing, machine learning, and advanced microprocessors. The opportunistic refresh port mode is suggested for maximizing memory utilization when a large number of ports is required, since it allows us to partially hide the refresh operations by utilizing temporarily unused ports during standard $NRMW$ operation. To demonstrate the proposed topology, an 8 kbit memory array based on a novel 4T bitcell offering 2R2W [15] capabilities was implemented in a 28 nm FD-SOI technology, providing between 1.8–3× area reduction compared to other dual-ported SRAM memories, while maintaining 100% memory availability.

*Contributions*

The major contributions of this paper are summarized as follows:

1. This work presents the first reported 2R2W and 6R2W GC-eDRAM memory, as well as a general design guideline for $N$-ported dynamic memories.
2. The resulting bitcell is the smallest multi-ported memory reported in the literature.
3. A novel opportunistic refresh port approach for dynamic memory arrays with $NRMW$ access ports is provided.
4. The proposed array can be dynamically configured to support an internal refresh operation without sacrificing memory availability.

The rest of this paper is structured as follows: Section 2 presents the proposed multi-ported circuit design and operating mechanism; implementation and extensive post-layout simulation results of the $NRMW$ cell are provided in Section 3; memory configurations and operating modes are presented in Section 4; Section 5 compares the proposed memory to other multi-ported memory options; and Section 6 concludes the paper.

## 2. Multi-Ported Gain Cell Design

### 2.1. 2R2W Cell Design and Operating Mechanism

A 2R2W gain-cell, designed with four transistors, is shown in Figure 4. This circuit is based on the general concept of the $NRMW$ GC-eDRAM cell design. As the least complex example of the proposed topology, this cell will be used as a reference design throughout the paper for demonstrating the memory properties. The circuit comprises two write ports featuring n-type MOSFET (NMOS) write access devices (NW1 and NW2) and two read ports, based on NMOS read access devices (NR1 and NR2). NMOS devices were preferred over p-type MOSFET (PMOS) devices due to faster access times [35]. In write mode, the write bit-line (WBL$_1$ or WBL$_2$) voltage is transferred to the storage node through its corresponding write access device (NW1 or NW2, respectively) when the appropriate write word-line signal (WWL$_1$ or WWL$_2$, respectively) is asserted. The write word lines are asserted by driving them to a boosted voltage (1.2 V) to enable passing a strong logic-'1' into the cell. The separation of the two write ports with independent write word-lines and write bit-lines provides the cell with 2W capability. During standby and read-only cycles, the write word-lines are discharged to −200 mV in order to suppresses the sub-threshold leakage from the storage node (SN) and extend the data retention time (DRT) [44].

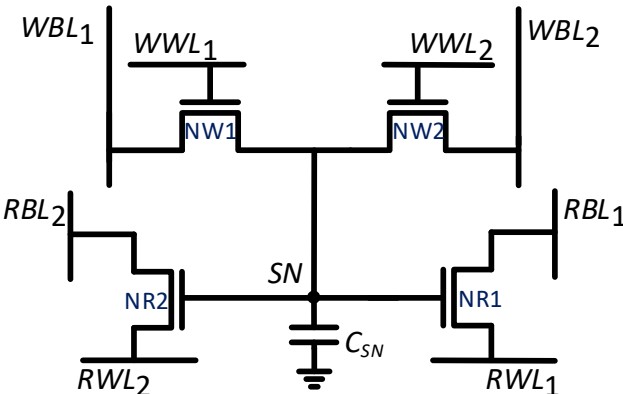

**Figure 4.** Schematic of the proposed 2R2W Gain Cell [15].

In read mode, the appropriate read bit-line ($RBL_1$ or $RBL_2$) is first pre-charged to $V_{DD}$ while the read word-line is kept closed (at $V_{DD}$). In the second phase of the read, its corresponding read word-line ($RWL_1$ or $RWL_2$, respectively) is discharged to GND. The corresponding read bit-line is then discharged if the cell is storing a logic-'1', or otherwise remains charged to represent a logic-'0'. Similar to write, the separation of the two read ports with independent read word-lines and read bit-lines provides the capability to achieve two parallel reads (2R) to complete the aforementioned 2R2W functionality.

The functionality of the proposed bitcell is demonstrated through the waveforms of Figure 5. The figure shows a simultaneous write operation to two bitcells at two different addresses, writing '0' to one cell and '1' to the second cell. These write operations are followed by simultaneous reads from the previously written addresses. It is clearly shown that $RBL_1$ is discharged to a value close to one threshold-voltage drop below the supply ($V_{DD} - V_T$), as its corresponding bitcell is storing a logic-'1', and similarly $RBL_2$ remains at the precharge level of $V_{DD}$, as its corresponding bitcell is storing a logic-'0'. In total, four independent rows of the array can be activated in parallel, enabling two separate write and two separate read operations to be performed simultaneously. When two conflicting (opposite) data values are written to the cell, the memory gives priority to port 1 ($WWL_1/WBL_1$), disabling the write access to the second port.

### 2.2. Expanding the Number of Read Ports

Multi-ported embedded memories with more than two read ports are even more beneficial for high-bandwidth applications [6]. Yet, due to large size and increased complexity of the SRAM implementations [5], these cells are uncommon [13]. Existing SRAM implementations are area costly and rely on decoupling the read ports from the cell itself, as shown in Figure 2a, for performing differential or single ended reads from those ports.

The proposed GC-eDRAM topology enables the design of an $(N + M)$-ported memory, using only $(N + M)$ transistors. For comparison, the eight-ported 6R2W memory, reported in [6], required 24 transistors, whereas the proposed topology achieves the same functionality using only 8 transistors. The same operating principle, demonstrated for the 2R2W cell, is maintained here. Every additional transistor is a standalone read or write port. Since the multi-ported read functionality is the more common requirement, in this work we focused on expanding the number of read ports. Nevertheless, expanding the number of write ports in the proposed cell design can also be achieved by adding a single transistor for each additional write port.

A cell with $N$ read ports will have $N$ read word lines, indexed from $RWL_1$ to $RWL_N$, with each word-line connected to the source of a corresponding read transistor, and $N$ read bitlines, indexed from $RBL_1$ to $RBL_N$, connected to the drain of the same corresponding read transistor. In a similar way, $M$ write port word lines are indexed $WWL_1$ to $WWL_M$ and are connected to corresponding write transistor gates, while $M$ write bitlines, indexed from $WBL_1$ to $WBL_M$, are connected to the drains of these write transistors. The proposed cell structure is presented in Figure 6. Due to the decoupled nature of GC-eDRAM circuits,

each write and read port operates independently, without affecting the operation of the other ports. When two conflicting (opposite) data values are written to the cell, the memory gives priority to port 1 (WWL$_1$/WBL$_1$). Due to the nature of gain cell design, in which the data is stored on the parasitic capacitance of the gates of the read transistors, simultaneous read of the same row by all ports is allowed and does not affect performance.

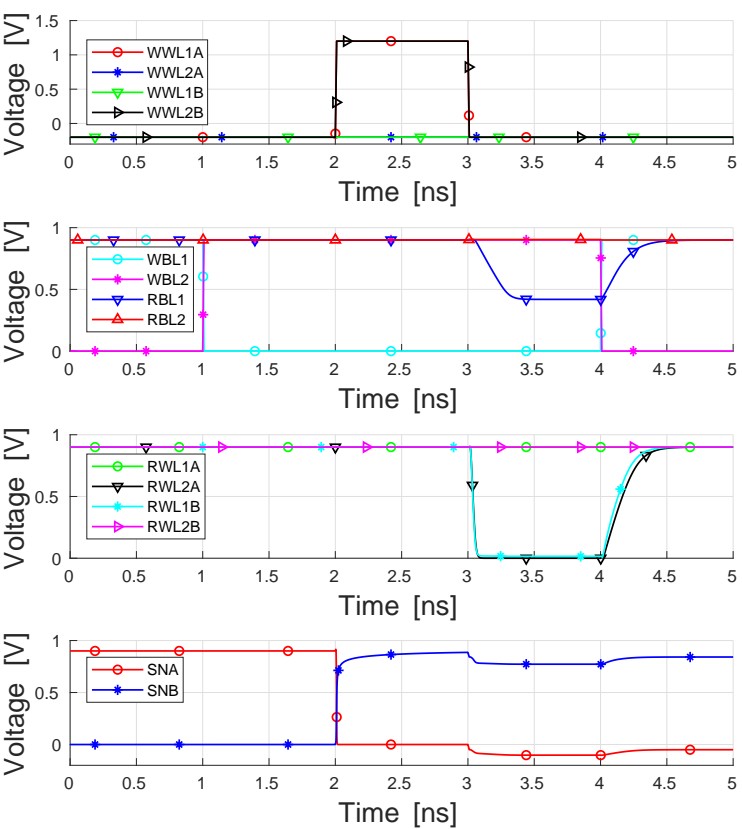

**Figure 5.** Waveform demonstration of two parallel write operations followed by two parallel read operations. Given four different rows, all of these four operations could have been performed during the same cycle [15].

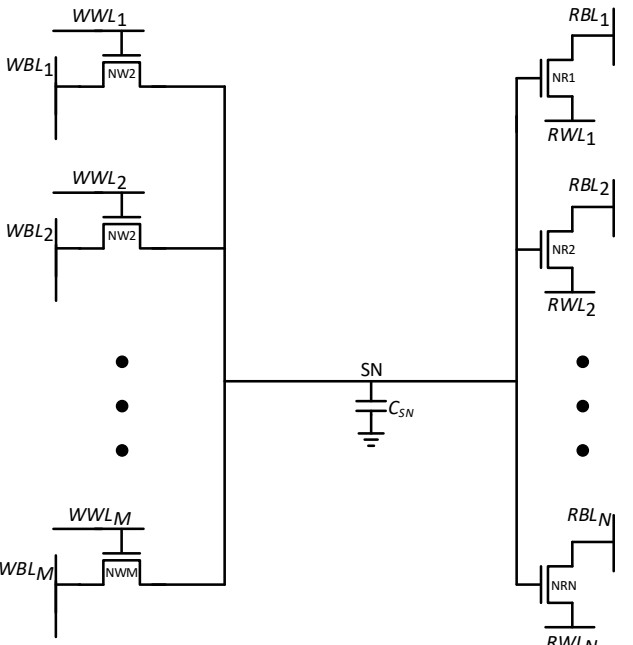

**Figure 6.** $N + M$ transistors are required to design an $NRMW$ ($N + M$-ported) GC-eDRAM bitcell.

## 3. Implementation and Simulation Results

In order to evaluate the proposed cell design characteristics, a representative 2R2W bitcell was implemented in a fully-depleted silicon-on-insulator (FD-SOI) technology, using regular-$V_T$ (RVT) NMOS transistors, which were preferred over low-$V_T$ (LVT) devices due to lower cell leakage. This section will overview the implementation and simulation results. Layout considerations for expanding the number of read ports are also described in this section.

### 3.1. 2R2W Bitcell Layout

Layout of the 2R2W bitcell is shown in Figure 7, measured at $0.24\,\mu m^2$ ($0.851\ \mu m \times 0.278\,\mu m$). The cell layout features WWL$_1$ and WWL$_2$ routed with horizontal *poly*  and *Metal-2* lines for reduced capacitance and better area utilization, respectively. RWL$_1$ and RWL$_2$ were routed with a horizontal *Metal-4* stripe, and the bit-lines were routed with a vertical *Metal-3* stripe. *Metal-2* is mainly used in cell routing. The access transistor connected to WWL$_2$ has a larger gate length than the access transistor connected to WWL$_1$ to reduce leakage. This port is also better for usage in the 1R1W mode for leakage reduction, as the write speed is not a limiting factor for the memory frequency [34].

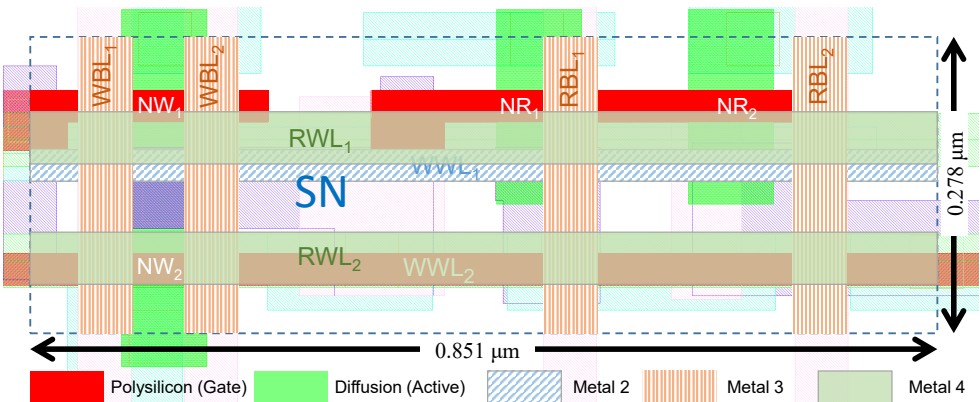

**Figure 7.** 2R2W bitcell layout in 28 nm FD-SOI [15].

### 3.2. Expanding the Layout to Accommodate Additional Read Ports

While any combination of read and write ports can be achieved in the proposed topology, the high density of the cell makes the signal routing the dominant consideration in the cell design. Additional cell area is required for routing the RWL signals, and higher metal layers are used to optimize the cell area utilization. Designing a cell with three additional read ports requires enlarging the height of the cell in order to accommodate the additional RWL metal stripes.

To demonstrate such a cell layout, the layout of an 8T-6R2W cell is illustrated in Figure 8, measured at $0.63\,\mu m^2$ ($1.773\,\mu m \times 0.357\,\mu m$). In order to add four additional read ports to the design, RWL$_1$, RWL$_2$, and RWL$_3$ were routed above the transistors, and RWL$_4$, RWL$_5$, and RWL$_6$ were routed through *Metal-4, 5, and 6* stripes at the bottom of the cell. These metal stripes require adding $0.079\,\mu m$ to the 2R2W cell height. The additional four read transistors enlarge the 2R2W cell width by $0.922\,\mu m$.

Note that, since *Metal-6* is often required for global power routing, avoiding its usage in the memory macro is possible at the cost of adding an additional $0.18\,\mu m^2$ to the cell area. Yet, as the GC-eDRAM is typically a self-contained block and the bitcells do not require power rail connections, routing through *Metal-6* may be acceptable for improving memory density.

The symmetric layout of the cell allows us to derive a formula for cell area (*A*) based on the number of read and write ports:

$$A(N,M)[\mu m^2] = \begin{cases} N + M \le 8, M \le 2: \\ \quad (0.278 + 0.079 \cdot \lfloor \frac{N}{4} \rfloor) \\ \quad \cdot (0.851 + (N-2) \cdot T_{hp}) \\ \\ N + M \le 8, M > 2: \\ \quad 0.477 \cdot (0.851 \\ \quad\quad + (N-2) \cdot T_{hp} \\ \quad\quad + (M-2) \cdot (T_{hp} + D_p)) \\ \\ N + M > 8: \\ \quad (0.357 + 2.2 \cdot W_{Mx} \\ \quad\quad\quad\quad \cdot \lfloor \frac{N+M-6}{3} \rfloor) \\ \quad \cdot (0.851 + (N-2) \cdot T_{hp} \\ \quad\quad + (M-2) \cdot (T_{hp} + D_p)) \end{cases} \tag{1}$$

where $W_{Mx}$ denotes the minimal metal width for the technology, defined as 0.05 µm for this process; $D_p$ is defined as the minimal distance between two poly stripes, measured at 0.096 µm for this process; and $T_{hp}$ is the horizontal pitch of minimal sized transistor for the given technology (including the gate tap), derived at 0.23 µm for this process.

The layout of Figure 8 shows how to route the signals of the added ports, according to the formula. While the upper third of the cell layout utilizes the routing tracks for in-cell connections, the middle and bottom thirds are primarily used to route the horizontal wordlines, going up to Metal-5. Each set of three wordlines can be routed across a single horizontal track by breaking the middle signal to allow space for vias that vertically connect the signals to the corresponding front-end connections. Following this approach, adding up to three additional wordlines requires extending the cell height by only 0.11 µm, and this can be repeated for any number of additional ports. Furthermore, each additional read transistor requires adding 0.23 µm to the cell width, and each additional write transistor requires adding 0.36 µm to the cell width. The difference between adding read and write transistors is derived from the constraint that write port poly cannot be shared and each gate needs its own tap, while for read transistors the poly is shared and they all connect to the same tap.

The numeric parameters in the formula are also technology dependent and calculated as follows: 0.278 µm is the minimum height that the two write transistors (sharing a diffusion) can fit; 0.851 µm is the width of the 2R2W cell comprised from $3T_{hp}$, $D_p$, and additional routing area; 0.079 µm is the added height for word lines 4–6 and derived from $W_{Mx}$. Yet, since the joined write transistors are wider than the metals required for word, the area added by word lines 4–6 is reduced.

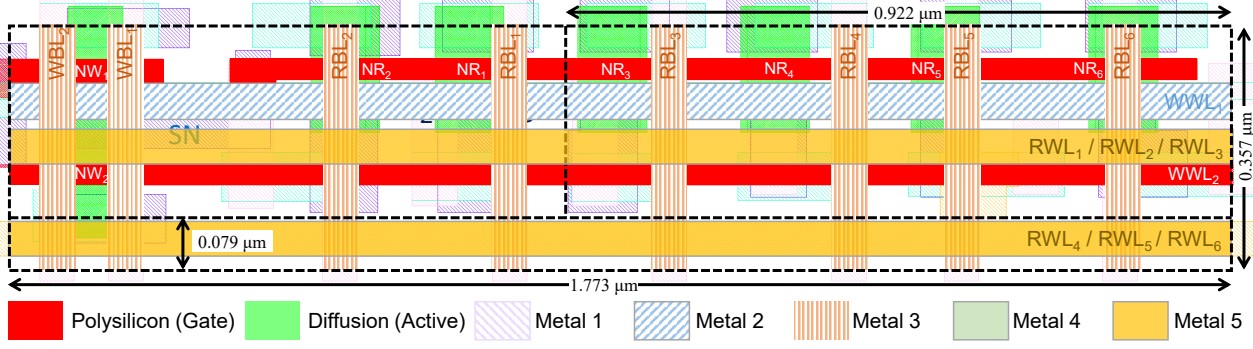

**Figure 8.** 8T-6R2W GC-eDRAM bitcell layout.

A comparison of the area of the proposed topology to other multi-ported designs is provided in Figure 9. The area of the multi-ported 6R2W bit-cell, illustrated in Figure 8, is only 1.9× bigger than the single ported 1WR SRAM cell, and when operating in the 1W5R mode, described in Section 4, it is 2.73× smaller than the corresponding SRAM cell described in [5].

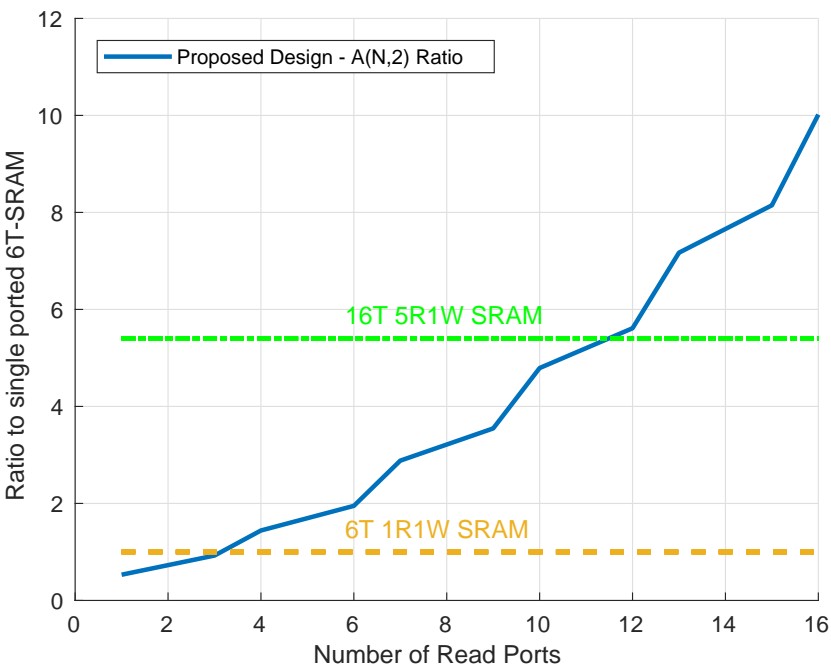

**Figure 9.** Area of the proposed design relative to a 6T-1RW GC-eDRAM cell.

### 3.3. Simulation Results

To quantify the characteristics of the proposed cell design, a 128 × 64 bit (8 kbit) memory macro, based on the reference 2R2W cell, was designed in a 28 nm FD-SOI process, and simulated under a 900 mV supply voltage at room temperature. All simulations were carried out with Cadence Virtuoso, using the Spectre circuit simulator, based on post-layout simulations applied to extracted views including parasitics. Retention time testbenches and estimations were performed according to the methodologies elaborated upon in [34].

In order to estimate the DRT of the implemented array, 1000 Monte Carlo (MC) statistical simulations, including mismatch and process variations, were applied to extract the deterioration of the storage node voltage following a write operation. The simulations were run under worst-case biasing conditions when operated in the 2R2W mode, i.e., $WBL_1$ and $WBL_2$ were biased to the opposite voltage of that stored on the SN to maximize the leakage current of the cell. The resulting deterioration curves of SN are shown in Figure 10, indicating a worst-case DRT of 6 μs, defined at the time when the difference between data '1' and '0' reaches beneath 400 mV.

To verify the readability of the bitcell, Figure 11 shows the distribution of the RBL voltage following read '0' and read '1' operations with a cycle time of 2 ns (500 MHz) following a 6 μs retention period. This plot was also extracted from 1000 post-layout MC simulations with mismatch modeling and including the layout-extracted parasitics. The simulations were run under worst-case conditions with all the unselected cells in the column storing a '1'. This disables the RBLs to discharge to '0' when reading '1' due to leakage from the unselected cells, causing the RBL to saturate slightly above 600 mV. Nevertheless, the voltage distributions of the RBL, when reading '1' and '0', are clearly separated, indicating that the array can be correctly read following the target retention period, with the switching threshold of the sense inverter positioned between the RBL distributions. The 6R2W multi-ported version of the bitcell demonstrates similar results.

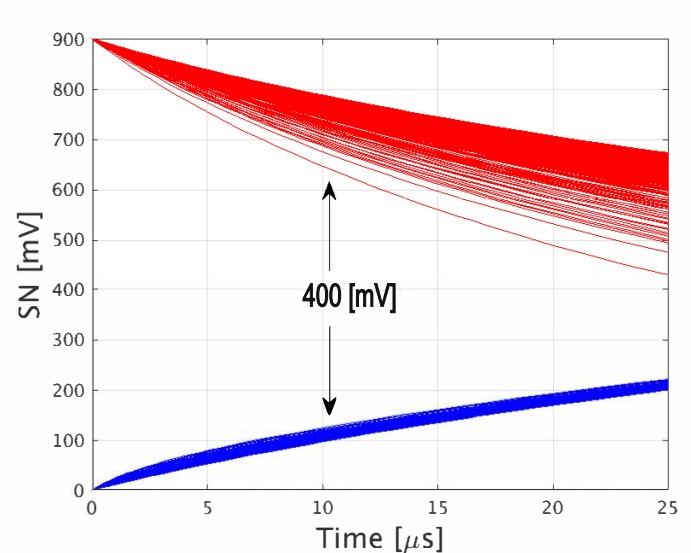

**Figure 10.** Storage node degradation of the proposed 2R2W gain cell following a write operation under worst case WBL bias conditions.

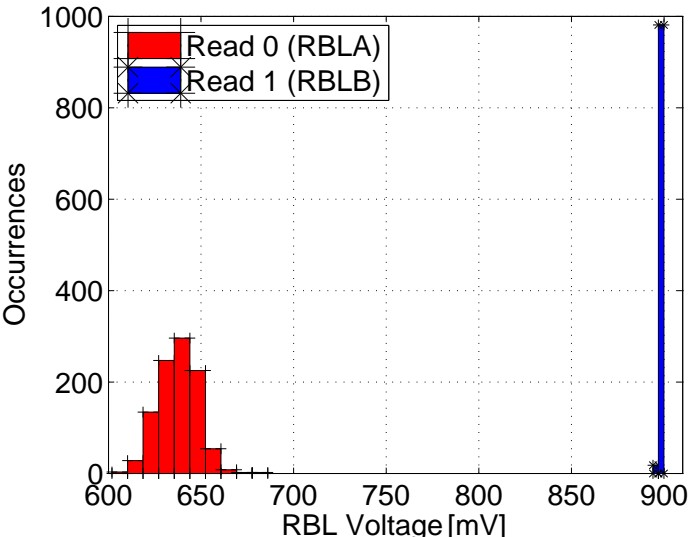

**Figure 11.** Distribution of the RBL voltage following read '1' and read '0' operations [15], while reading from two different ports through RBLA and RBLB (at different rows) simultaneously.

The performance of the read ports is exactly the same as in the 2R2W version of the cell, since they are completely independent of each other. The write ports are slightly slower due to the increased SN capacitance, yet since the write operation is much faster than the read, as illustrated in Figure 5, the overall performance is unchanged. Retention time on the other hand is increased as we add additional read or write ports, as illustrated in Figure 12. For example, increasing the number of read ports from six to sixteen yields an almost 2× retention time improvement. In terms of SN capacitance, increasing the number of write ports from two to twelve will result in an increase from 0.5 fF to 1.6 fF. Increasing the number of read ports is more efficient in terms of retention time since we increase the SN capacitance by adding poly-silicon area, at a cost of a relatively small gate leakage. When we increase the number of write ports, in addition to the increased storage node capacitance, we also have additional leakage sources through the added write transistor gates. Overall, the access performance of the suggested cell design does not change as a result of increasing the number of ports, while the increased retention time for a large number of ports improves the memory availability.

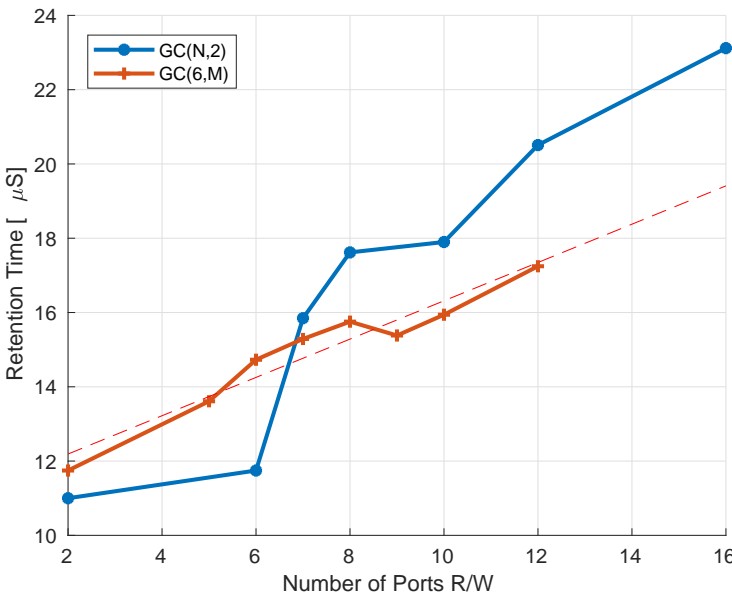

**Figure 12.** Retention time as a function of the number of ports. $GC(N, 2)$ demonstrates an increase in DRT as a function of the number of read ports, when the number of write ports is kept constant. $GC(6, M)$ shows DRT increase as a function of the number of write ports. Due to layout considerations derived from (1), a single port increase is not always beneficial, yet the DRT improvement tendency is clearly shown.

The leakage power of the 2R2W cell, simulated under worst-case biasing conditions, was found to be $0.93$ pW/bit and $1.2$ pW/bit for the 6R2W cell. The worst case biasing conditions are different between the cells since for the 6R2W bitcell, the read ports gate leakage becomes more substantial than the write ports gate leakage. The active refresh energy was found to be $65$ fJ/bit, composed of $25$ fJ/bit for read and $40$ fJ/bit for write.

## 4. Configurable Operation

The previous sections demonstrated that the proposed bitcell can be operated as a *NRMW* bitcell, with *N* read ports and *M* write ports, focusing on a detailed evaluation of the 2R2W implementation of this topology, and the implications of adding additional read and write ports. In this section, we will show how the bitcell can be dynamically configured for optimization of the system memory usage.

### 4.1. NRMW Operation Mode

The standard operation mode of the proposed topology, as presented earlier, provides *NRMW* functionality for high-bandwidth applications that require multi-ported operation, where as many as *N* simultaneous reads and *M* simultaneous writes (*NRMW*) are needed. This is achieved by utilizing each pair of write word-lines and write bit-lines ($WWL_1/WBL_1$, $WWL_2/WBL_2$ up to $WWL_M/WBL_M$) for write operations, and each pair of read word-lines and read bit-lines ($RWL_1/RBL_1$, $RWL_2/RBL_2$ up to $RWL_N/RBL_N$) for read operations. While $N + M$ separate row-decoders are required to support this extensive functionality, the requirement can be reduced to fewer row decoders. For example, in regards to the 4T cell presented in Section 2.1, if dual-ported (2RW) operation is sufficient, the number of decoders can be reduced to two decoders. As mentioned before, there is no conflict between the ports and all the ports can operate simultaneously without a performance penalty. When operated in *NRMW* mode, the bitcell may still require periodic refresh operations, yet for the suggested design, $\min(N, M)$ ports can be used in parallel to perform the refresh, at the cost of memory blockage for $(R + 1)/\min(N, M)$ clock cycles, where $R$ is the number of rows in the array. This improves the memory availability for arrays with a large number of access ports.

### 4.2. Internal Refresh Mode

Dynamic memories, such as GC-eDRAM, suffer from limited DRT due to leakage currents through the storage node. The deterioration of the data levels under worst-case biasing conditions is shown in Figure 10, with the DRT defined at the point where the logic-'1' (red lines) and logic-'0' (blue lines) intersect. In order to ensure data integrity, refresh operations, which read out and write back all the data in the memory array, are periodically applied. In addition to the refresh power consumption, the memory availability is compromised, since the memory is unavailable for external accesses when refresh is applied. This would render the proposed memory incompatible with certain applications that cannot tolerate such occurrences.

To address this issue, the proposed design includes a unique configuration capability, which can provide inherent 100% array availability at the expense of a reduction in the number of available ports. In this mode, the $NRMW$ capability is reduced—at least temporarily—to $(N-1)\mathrm{R}(M-1)\mathrm{W}$, and the other ports are used for hidden refresh of the array. This loss of capability can be tolerated by applications that do not need the full $NRMW$ functionality, while still providing at least a two-ported (1R1W) memory that can be used as a direct replacement for SRAM with both area and power reductions, while avoiding the need to integrate special controllers or dedicated refresh-aware protocols. These kinds of techniques are often applied for DRAM memories, where the many-word rows and longer read latency allow even further customization. These techniques are often referred to as parallel and hidden refresh, where the refresh operation is hidden inside the DRAM controller by design [45–49].

To apply the hidden refresh mode, low indexed ports $WWL_1/WBL_1$ and $RWL_1/RBL_1$ up to $WWL_{M-1}/WBL_{M-1}$ and $RWL_{N-1}/RBL_{N-1}$ are used to perform the standard write and read operations, respectively, as requested by the system. A refresh controller that is provided along with the memory macro and transparent to the outer architecture uses the last set of ports, $WWL_M/WBL_M$ and $RWL_N/RBL_N$, to perform the refresh operation. In this way, the refresh is hidden from the outer system and applied without interfering with standard multi-ported array accesses. The area overhead of the dedicated refresh ports is almost negligible for cells with a large number of ports. For example, when implementing a 10R8W bitcell, the dedicated refresh ports will consume only 11% of the cell area. Note that if the system is not required to dynamically reconfigure the memory, the last set of address decoders can be replaced by a simple shift-register-based state machine to carry out the refresh operations. In addition, the DRT of the array is slightly higher than in $NRMW$ mode, as the bias value can be controlled during non-refresh cycles and set at the best-case level.

### 4.3. High Performance Opportunistic Refresh Port Mode

The multi-ported nature of the $NRMW$ bitcell allows us to combine the multi-ported refresh approach, explained in Section 4.1, with the hidden refresh mode to enable an *opportunistic refresh port* approach. This approach dynamically chooses a read or write port that is not being used by the system in the current cycle, and uses it as a read or write port for refresh. In this mode, instead of completely blocking the ports used for refresh, the memory controller waits until one of the ports becomes available (i.e., not used by the system), and performs a refresh operation using this port. This technique reduces the percentage of time the memory is blocked for read or write, from $(\alpha_r + \alpha_w)$ of opportunistic refresh algorithm such as [50], down to $(\alpha_r/N + \alpha_w/M)$, where $\alpha_r$ and $\alpha_w$ denote the memory refresh read and write blockage during normal system operation. For the case of the 2R2W cell (the minimal proposed configuration), the availability is improved by 50%.

The suggested refresh controller operation is described in Algorithm 1. It is based on using an available read port for reading the refreshed row into a buffer, and later—when there is an available write port—writing it back. The main principle of the suggested algorithm is to use any free port it can find for the refresh operation. The operations on read and write ports are performed in parallel (denoted by "⇒") as all the ports are completely

independent. While the write port algorithm is a straightforward search for an available write port to use for the refresh, the read ports operation is more complex. Since a single buffer is used both for storing the read data and writing from it, it can be read into in only one of the following two cases:

1. The previous data is written back into the array at the current cycle.
2. There is no data in the buffer.

---

**Algorithm 1** Refresh Controller Algorithm

---

**for** ever **do**
  // Wait until next refresh
  $cycles = 0$
  **while** $cycles < DRT\_cycles \cdot SE\_Factor$ **do**
    $cycles + +$
  // Perform opportunistic refresh
  $read\_row = 0$
  $write\_row = 0$
  **while** $write\_row < N_{ROWS}$ **do**
    ⇒**if** *data stored in buffer* **do**
      **for** each in *WRITE_PORTS* **do**
        **if** *PORT* is *free* **then**
          **write** *data* from *buffer* to *write_row*
          $write\_row + +$
          break
    ⇒**for** each in *READ PORTS* **do**
      **if** *PORT* is *free* **then**
        **if** *buffer* is empty **or** *RW_cycle* **then**
          **read** *data* from *read_row* to *buffer*
          $read\_row + +$
        break
    $cycles + +$
    **if not** $cycles < DRT\_cycles$ **then**
      break
  // Check opportunistic refresh success
  **if** $write\_row \neq N_{ROWS}$ **then**
    initiate blocking refresh

---

Since read access time is typically the frequency limiting factor, in case there is data stored in the buffer, we do not want to wait for the logic that checks for write port availability before searching for a free read port. Instead, we look for a free read port in parallel to looking for a free write port, and read the new data only if the buffer is empty or the stored data is written in the current cycle (i.e., a free write port is found, denoted by *RW_cycle*). In case the controller fails to complete the refresh in the time slot given to the *opportunistic refresh*, a blocking sequential refresh is enforced on the system, in a similar way to other opportunistic algorithms [50–53].

This refresh scheme can be extremely useful for systems in which different devices are connected to different ports, such as the architecture suggested in [7] where the memory is connected in up to six parallel execution units. In such a configuration, there will be gaps in memory utilization at the system level, during which a certain peripheral does not use the memory. This time can be used as an opportunity for the memory controller to perform refresh operations without disturbing the system, almost completely eliminating memory blockages. An example of such a system configuration is illustrated in Figure 13, where the suggested memory is implemented for a 2W6R configuration. For this design, we assume that one accelerator (on the right) requires one write port and two read ports, and the other accelerator (on the left) requires a single read port. The controller detects cycles when there is no access from an external interface to one of the ports and internally

utilizes this memory port to perform the required refresh operation. It should be noted that for configurations with several write ports, the controller has to keep track to ensure there are no write collisions and that the row in which data is stored in the *buffer* has not been overwritten since the read. In case the row is overwritten by external access, the *buffer* should be discarded and the next row should be read.

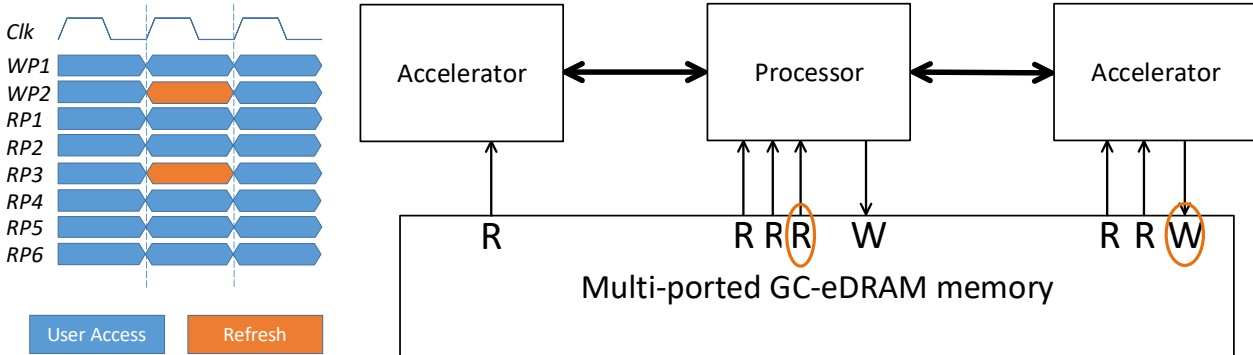

**Figure 13.** System design with main processor, two accelerators, and the implementation for a 2W6R configuration with the suggested controlling algorithm. For a cycle in which the write port of one of the accelerators and the main processor's read port are unused, they can be utilized for performing the refresh operations. The controller can use any free ports from the same or different interfaces for the refresh.

## 5. Comparison to SRAM

Table 1 shows a comparison between the proposed 4T 2R2W bitcell and other multi-ported embedded memory options, including a 2T gain-cell [34], 6T SRAM [54], 8T SRAM [3], and a 10T SRAM [2,55]. A major advantage of the proposed cell over other dual-ported SRAM implementations is in the cell area size, offering between a 1.8–3× reduction over the compared SRAM implementations with the same number of ports. While the 2T gain-cell offers the lowest area, it only provides two-ported (1R1W) operation, as compared to the 2R2W functionality offered by the smallest configuration of the proposed topology. Moreover, while the 2T gain-cell provides almost the same DRT as the proposed 2R2W cell, it has a reduced memory availability of 97.3% due to the refresh operation occupying the single write and read ports. Alternatively, if operated in the internal-refresh (1R1W) mode, the proposed topology provides 100% availability. This is a significant advantage over standard GC-eDRAM offerings, as it simplifies the system design and enables straightforward replacement of SRAM with GC-eDRAM memories.

Table 2 compares the many-ported implementations. Three different bitcells based on the suggested design are compared with SRAM-based options. The first comparison is for a 5R1W bitcell that is compared with a 16T SRAM-based bitcell [5]. The second comparison is between 8R1W implementations; our implementation compared to a 20T SRAM-based bitcell [4]. Finally, a 12T 6R6W Gain Cell is compared with the 16T 3W3R SRAM-based bitcell, which is pumped to 6R6W functionality in [16]. Even though the GC-eDRAM bitcell is compared with a bitcell that has half the number of ports, the GC-eDRAM area savings are significant. This can be explained by the symmetrical and NMOS-only based design of the GC-eDRAM memory. Moreover, techniques such as double-pumping are rarely applicable, since the memory itself is often the performance bottleneck in the system, and if the memory can go faster, it is more beneficial to activate the whole system at a higher frequency.

The area advantage becomes even more substantial when comparing an *NRMW* configuration to other multi-ported offerings. A 6R2W configuration of the proposed topology is only 1.9× larger than a standard 1RW SRAM cell. When compared to a cell with similar functionality, the proposed 6R2W configuration in internal-refresh mode provides 5R1W functionality, while being 2.8× smaller than the 5R1W SRAM cell of [5]. Moreover, while multi-ported SRAM requires special techniques to maintain sufficient noise margins during read operations (e.g., [6]), the decoupled read of the proposed GC-eDRAM

multi-ported configuration allows for adding any number of read ports for simultaneous operation. For a larger number of ports, the improved DRT and availability completely eliminate the disadvantages of the GC-eDRAM, making it a superior multi-ported solution over SRAM.

**Table 1.** Comparison between the proposed 4T design and other similar SRAM-based memory options with a small number of ports.

| | 6T SRAM [54] | 8T 1W1R SRAM [54] | 8T 2WR SRAM [3] | 10T 1W2R SRAM [2] | Conventional 2T-NMOS Gain-Cell [34] | Proposed 4T Gain-Cell [This Work] |
|---|---|---|---|---|---|---|
| Technology Node | 28 nm FD-SOI | 28 nm FD-SOI | 65 nm CMOS | 45 nm CMOS | 28 nm FD-SOI | 28 nm FD-SOI |
| Supply Voltage | 0.7 V | 0.48–0.7 V | 1.2 V | 1 V | 0.9 V | 0.9 V |
| Availability | 100% | 100% | 100% | 100% | 97.3% * | 100% |
| Cell Size | 0.325 $\mu m^2$ | 0.42 $\mu m^2$ | 0.71 $\mu m^2$ | 0.8 $\mu m^2$ | 0.152 $\mu m^2$ | 0.23 $\mu m^2$ |
| Ratio to 6TSRAM | 1 X | 1.3 X | 1.44 X | 2.14 X | 0.47 X | 0.71 X |
| Data Retention Time | Static | Static | Static | Static | 9.6 $\mu s$ | 11 $\mu s$ |
| Leakage Power (In 28 nm FD-SOI) | 12.9 pW/bit | 16.3 pW/bit | 13.1 pW/bit | 16.7 pW/bit | 576 fW/bit | 1.25 pW/bit |

All cells were simulated under a 900 mV supply at 27 C. * Assuming a 2 ns cycle and a 128-row memory.

**Table 2.** Comparison between the proposed design for large ports amount and other memory options.

| | 6T 1W5R Gain Cell [This Work] | 16T 1W5R SRAM [5] | 9T 1W8R Gain Cell [This Work] | 20T 1W8R SRAM [4] | 12T 6W6R Gain Cell [This Work] | 16T 3W3R/6W6R SRAM [16] |
|---|---|---|---|---|---|---|
| Technology Node | 28 nm FD-SOI | 90 nm CMOS | 28 nm FD-SOI | 40 nm CMOS | 28 nm FD-SOI | 7 nm CMOS |
| Supply Voltage | 0.9 V | Not Reported | 0.9 V | 1.1 V | 0.9 V | 0.9 V |
| Availability * | 97.3% * | 100% | 97.3% * | 100% | 99.1% * | 100% |
| Cell Size | 0.55 $\mu m^2$ | 7.35 $\mu m^2$ | 1.04 $\mu m^2$ | 3.12 $\mu m^2$ | 1.78 $\mu m^2$ | 0.39 $\mu m^2$ |
| Ratio to 6TSRAM | 1.7 X | 7.36 X | 3.2 X | 12.9 X | 5.46 X | 14.25 X |
| Data Retention Time | 21.8 $\mu s$ | Static | 23.5 $\mu s$ | Static | 15.2 $\mu s$ | Static |
| Leakage Power (In 28 nm FD-SOI) | 3.12 pW/bit | 34.6 pW/bit | 3.45 pW/bit | 90 pW/bit | 12.6 pW/bit | 108 pW/bit |

* For NRMW mode with blocking refresh, assuming a 2 ns cycle and a 128-row memory.

## 6. Conclusions

This paper proposed a novel multi-ported gain-cell eDRAM bitcell topology that provides up to $N$ simultaneous read and $M$ simultaneous write ($NRMW$) operations for use in high-bandwidth applications that require several memory operations in parallel. The configuration of some of these ports for applying internal refresh provides the opportunity to use the proposed topology as a straightforward SRAM replacement with 100% memory availability. By further exploiting the unused ports for opportunistic refresh, the memory availability can be maximized while maintaining near $NRMW$ functionality. Compared to other dual-ported SRAM memory options, the proposed 2R2W cell provides between 1.8–3× area reduction and the lowest leakage power consumption, providing much higher savings as compared to memories with more ports. Compared to conventional GC-eDRAM memories, the multi-ported cell configured for internal-refresh offers 100% memory availability, overcoming one of the main drawbacks of dynamic memory solutions.

**Author Contributions:** Conceptualization, R.G. (Roman Golman) and R.G. (Robert Giterman); analysis, simulation, and writing, R.G. (Roman Golman); supervision, review and editing, A.T. All authors have read and agreed to the published version of the manuscript.

**Funding:** This work was funded by the Israel Ministry of Science, Innovation and Technology (MOST) under the the Lise Meitner Grants for Israeli-Swedish Research Collaboration Project "AutoPiM".

**Data Availability Statement:** Restrictions apply to the availability of these data. Data was based on intellectual property of third parties accessed under non-disclosure agreements.

**Conflicts of Interest:** The authors declare no conflict of interest.

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
