# Peer review of "Multi-Ported GC-eDRAM Bitcell with Dynamic Port Configuration and Refresh Mechanism"

_jlpea, doi:10.3390/jlpea14010002_

Round 1

Reviewer 1 Report

Comments and Suggestions for Authors

This paper proposed a novel multi-ported gain-cell eDRAM bitcell topology that provides up-to N simultaneous read and M simultaneous write (NRMW) operations for use in high-bandwidth applications that require several memory operations in paralle. Here are some suggestions for paper improvement:

1.The author should create a table that highlights the innovation of this article by comparing multiple indicators.

2.The references do not cover the latest work in the past two years.
3.The author did not elaborate in the introduction on the design concept of the "a novel multi ported gain cell eDRAM bitcell topology" circuit proposed in this article, as well as its background in engineering applications.

4.In, Fig. 12, The author only provided a simple explanation and provided a module without providing detailed explanations.

5. In Fig. 2, The layout is quite chaotic and should be further improved to make it easier for readers to understand

Comments on the Quality of English Language

No comments.

Reviewer 2 Report

Comments and Suggestions for Authors

Summary:

Area and power of memory is the most dominant candidate for optimization in the modern embedded system design. Memory is accessed by multiple units simultaneously, which introduces a multi-ported memory design. However, the multi-port design increases access latency and area and reduces port access efficiency due to the refreshing mechanism. This article proposes a configurable multi-port gain cell (N-read port and M-write port - NRMW) with N+M transistors (NMOS) that intelligently refresh memory whenever possible or whenever the free ports are available.

The design was evaluated with the designed 2R2W (2-read ports, 2-write ports with 4-NMOS transistors) and 6R2W (6-read ports, 2-write ports with 8-NMOS transistors). One of the common complicated situations in multi-ported memory is appearing multiple writes in the same memory cell at the same time. This conflicting situation was resolved by prioritizing the first write port. However, simultaneous reads to the same cell are possible due to the nature of GC design, which states the data is stored on the parasitic capacitance of the gate of read transistors.

This design is evaluated by implementing 2R2W in an FD-SOI technology. The operating voltage was below 900-mV, in 28-nm process technology. Estimated the data retention time (DRT) with a 1000 Monte Carlo (MC) statistical simulation methodology.

Strength:

This design reduces 1.8-3x of bit-cell area compared to other dual-ported SRAM and provides 100% memory availability as opposed to conventional dynamic memory in most cases. One of the important findings is the DRT improved with the increase in read-write ports due to an increase in SN capacitance. To keep the memory array available even at the refresh time, the authors propose a unique approach where they dedicate a port to refresh the array at runtime. So, anytime at least (N-1) read ports and (M-1) write ports are available for service. Using a dedicated port at runtime refresh removes the necessity of extra protocols. However, an opportunistic refresh mechanism has also been proposed where the memory controller determines an idle port to use for refreshing.

Needs clarification:

Explanation of the memory layout with ports in more detail. The modeling tools and test sets have not been explained clearly. However, the authors have used Monte Carlo statistical simulation but did not explain the reliability analysis clearly. A comparison in leakage power, SN cap., refresh energy, DRT, between 2R2W/6R2W (implemented in GC-eDRAM) and any other proposed design (8T, 6T, 1R1W, 1T1C eDRAM etc.) should be shown here.

Comments on the Quality of English Language

Use some of the available tools to proofread and ensure no mistakes exist.

Round 2

Reviewer 2 Report

Comments and Suggestions for Authors

N/A